# De novo *NIPBL* Mutations in Vietnamese Patients with Cornelia de Lange Syndrome

**DOI:** 10.3390/medicina56020076

**Published:** 2020-02-14

**Authors:** Duong Chi Thanh, Can Thi Bich Ngoc, Ngoc-Lan Nguyen, Chi Dung Vu, Nguyen Van Tung, Huy Hoang Nguyen

**Affiliations:** 1Institute of Genome Research, Vietnam Academy of Science and Technology, 18 Hoang Quoc Viet str., Cau Giay, Hanoi 100000, Vietnam; mr.chithanhduong@gmail.com (D.C.T.); lannguyen@igr.ac.vn (N.-L.N.); tungnv53@gmail.com (N.V.T.); 2Center for Rare Diseases and Newborn Screening, Department of Endocrinology, Metabolism and Genetics, Vietnam National Hospital of Pediatrics, 18/879 La Thanh str., Dong Da, Hanoi 100000, Vietnam; bscanbichngoc@gmail.com (C.T.B.N.); dungvu@nch.org.vn (C.D.V.)

**Keywords:** Cornelia de Lange, de novo mutation, *NIBPL*, c.6697G>A, c.2602C>T, c.4504delG, whole exome sequencing

## Abstract

Cornelia de Lange Syndrome (CdLS) is a rare congenital genetic disease causing abnormal unique facial phenotypes, several defects in organs and body parts, and mental disorder or intellectual disorder traits. Main causes of CdLS have been reported as variants in cohesin complex genes, in which mutations in the *NIPBL* gene have been estimated to account for up to 80%. Our study included three Vietnamese patients with typical CdLS phenotypes. Whole exome sequencing revealed two known heterozygous mutations c.6697G>A (p.Val2233Met) and c.2602C>T (p.Arg868X), and a novel heterozygous mutation c.4504delG (p.Val1502fsX87) in the *NIPBL* gene of the three patients. In silico analyses of the identified mutations predicted possible damaging and truncating effects on the NIPBL protein. Inherited analyses in the patients’ families showed that all of the mutations are de novo. Our results lead a definitive diagnosis of patients with CdLS and expand the spectrum of mutations in the *NIPBL* gene. These findings also confirm whole exome sequencing is an efficient tool for genetic screening of CdLS.

## 1. Introduction

Cornelia de Lange Syndrome (CdLS; OMIM #122470, 300590, 3000882, 610759, and 614701) is a congenital genetic disorder that affects both mentally and physically aspects of patients [1,2]. The diagnosis of CdLS cases has been crucially relied on craniofacial anomalies which include synophrys and/or thick eyebrows; short nose, concave nasal ridge and/or upturned nasal tip; long and/or smooth philtrum; thin upper lip vermilion and/or downturned corners of mouth [3,4]. Nonfacial cardinal CdLS phenotype includes hand oligodactyly and/or adactyly and congenital diaphragmatic hernia [3]. Mild or non-classical CdLS may lack some of the pivotal facial and non-facial cardinal CdLS features and present some suggestive features such as global developmental delay and/or intellectual disability, prenatal growth retardation (<2 SD), postnatal growth retardation (<2 SD), microcephaly (prenatally and/or postnatally), small hands and/or feet, short fifth finger, and hirsutism depended on the case’s magnitude [3,5,6,7]. The prevalence of CdLS is ranging from 1 in 10,000 to 1 in 30,000 live births [8].

As the molecular technology advances, the main aetiology of CdLS has been elucidated as variants in genes encoding cohesin subunits (*SMC1A*, *SMC3*, and *RAD21*) and regulators (*NIPBL* and *HDAC8*) [1,7,9,10]. Cohesin is an essential protein complex for the chromosome segregation in dividing cells, DNA repair mechanism, and gene regulations [11,12]. Close to 70–80% of CdLS cases are associated with the mutations in the *NIPBL* gene [3,13]. The *NIPBL* gene encodes for delangin, an ortholog of Scc2 protein in *Drosophila*, which, coupled with Scc4 protein, regulates the loading of the cohesin ring onto chromatin [1,11]. Other identified variants related to CdLS occurred in the *SMC1A*, *HDAC8*, *SMC3,* and *RAD21* genes with less than 5% frequency [10]. Rad21 is one *Schizosaccharomyces pombe* ortholog of Scc1, a key protein in DNA damage repair process [11]. The *SMC1A*, *SMC3,* and *RAD21* genes encode cohesin subunit proteins which form a ring-like structure with chromatid entrapping functions. HDAC8 protein deacetylates SMC3 to regulate the release of the cohesin from the sister chromatids [14]. Mutations in *NIPBL*, *RAD21,* and *SMC3* genes are often reported as autosomal dominant while mutations in *HDAC8* and *SMC1A* genes are inherited in an X-linked manner [15]. Besides cohesin associated mutations, variants in *ANKRD11* gene have been presented in cases of non-classical CdLS phenotypes which overlap with KBG syndrome and Coffin-Siris-like syndrome phenotypes [16]. Deletion and variants in the *BRD4* gene have been reported in atypical CdLS patients [17]. In addition, variants in *EP300*, *AFF4*, *NAA10*, and *TAF6* genes have been identified in patients who showed features that overlap with CdLS [3].

In this study, we performed whole exome sequencing in three Vietnamese patients with CdLS. Three mutations in the *NIPBL* gene, which include two known heterozygous mutations c.6697G>A (p.Val2233Met) and c.2602C>T (p.Arg868X), and a novel heterozygous mutation c.4504delG (p.Val1502fsX87) have been assessed as potential causes of CdLS in the three patients. Sanger sequencing in their parents revealed the mutations as de novo in the patients.

## 2. Presentation of Case Reports

This study was conducted in accordance with the Declaration of Helsinki, and the protocol was approved by the Ethics Committee of the Institute of Genome Research (No. 18/QD-NCHG on 22 March, 2018, Institute of Genome Research Institutional Review Board, Hanoi, Vietnam). Three patients were included in this study. The informed written consent form was obtained from their parents. All patients (coded name CDL1, CDL2, and CDL3) and their parents were checked at Vietnam National Children’s Hospital.

Patient CDL1 is the first child of nonconsanguineous Vietnamese parents. He was born after 38 weeks gestation by normal delivery with a birth weight of 2.5 kg (−3 SD). At 3 months of age, he presented prolonged vomiting, developmental delay, and attention deficit/hyperactivity disorder. He could walk at 16 months of age. He was referred to the genetics clinic due to developmental delay. At 18 months of age, the patient had a weight of 7.8 kg (−3 SD), height of 73 cm (−3 SD), head circumference of 43 cm (<−2 SD), and developmental quotient of 60%. He presented a dysmorphic face with confluent eyebrows, synophrys, long eyelashes, thin upper lip vermilion, downturned corners of the mouth, long philtrum, low anterior hairline, low-set ears, and high arched palate (Figure 1(a1)). He was marked with hirsutism, partial syndactyly of the left second and third toes, clinodactyly of the fifth fingers, limitation of elbow extension, and smallness of hands and feet (Figure 1(a2,a3)). Brain magnetic resonance imaging revealed delayed myelination. He had normal renal and liver functions.

Patient CDL2 is the first child of nonconsanguineous Vietnamese parents. He was born after 36 weeks gestation by normal delivery a birth weight of 1.7 kg. He was referred to the genetics clinic at the age of two and a half years due to psychomotor retardation and convulsion but no fever. He was diagnosed with epilepsy and treated with depakin and keppra. His weight and height were 8 kg and 80 cm, respectively. His head circumference was 49.5 cm (+0.69 SD). He had dysmorphic facial features, including arched eyebrows, synophrys, short nose with concave nasal ridge, long philtrum, thin upper lip vermilion, and downturned corners of the mouth (Figure 1(b1)). He was marked with short fifth finger of the right hand, and smallness of hands and feet (Figure 1(b2,b3)). He could roll over but could not sit or speech. His penis was 2 cm long with hypospadias. His left testis was in the left scrotum and no right testis was presented in right scrotum.

Patient CDL3 was born full term after a normal delivery. He has been evaluated for mental and motor development delay. At 17 months of age, his height was 67 cm (−5.3 SD), his weight was 5.9 kg (≤3 SD) and his head circumference was 38 cm (−7 SD). He presented with a dysmorphic face (arched eyebrows, synophrys, short nose with concave nasal ridge, long philtrum, and thin upper lip vermilion with down-turned corners of the mouth) (Figure 1(c1)), clinodactyly of the fifth finger on the right hand and abnormal palmar crease on the left hand (Figure 1(c2,c3)). He could roll over at 15 months of age but could not sit.

All the parents in three cases were completely healthy with no positive history of any deformity in their respective families. Clinical scores of each patient were given according to the guidelines suggested in the first international consensus statement of CdLS [3].

Applying the guidelines suggested in the first international consensus statement of CdLS [3], our three patients CLD1, CDL2, and CDL3 received a clinical score of 12, 12, and 13, respectively (Table 1). All three patients were categorized as classical CdLS since the clinical score was greater than 11.

Three mutations in exon 10, exon 21, and exon 39 of the *NIPBL* gene were detected in three CdLS patients by whole exome sequencing and verified by Sanger sequencing (Figure 2). None of these mutations were detected in the in house database of 44 non-CdLS patients.

In patient CDL1, a heterozygous missense mutation c.6697G>A was detected (Table 1). This mutation is located in exon 39 of the *NIPBL* gene and involves a change from valine to methionine at residue 2233 (p.Val2233Met) on the delangin protein (Figure 2a,b). The mutation was previously reported in the ClinVar database (ID: VCV000644755.1) and a previous report [18]. Val2233 is conserved in all species examined (Figure 3b). With a sorting intolerant from tolerant (SIFT) score of 0.002 and polymorphism phenotyping version 2 (PolyPhen-2) score of 0.999, the mutation is predicted to be deleterious (Table 1).

Patient CDL2 carries a novel heterozygous single G deletion (c.4504delG) in exon 21 of the *NIPBL* gene (Table 1 and Figure 2a,c). The mutation substituted valine at the residue 1502 with tyrosine (p.Val1502TyrfsX87), resulting in a frameshift which truncated the protein 87 amino acids downstream (Figure 2c).

In case of patient CDL3, a heterozygous mutation c.2602C>T (p.Arg868X) was detected in exon 10 of the *NIPBL* gene (Table 1 and Figure 2a). The replacement of arginine with a stop codon lead to an early termination of the NIPBL protein at residue 868 (Figure 2d). This substitution was reported in ClinVar database (ID: VCV000096337.2) as a pathogenic mutation and in the dbSNP database (ID: rs398124466) [19].

All the detected mutations were de novo as they were absent in the parents (Figure 2b–d).

## 3. Discussion

Our study consists of three Vietnamese patients with CdLS. Due to the diversity of CdLS clinical presentations, Kline et al. recommended a scoring system which assesses a case as classical CdLS if the patients exhibit at least three cardinal features and achieve a total clinical scores equal to or more than 11 [3]. All our patients presented at least three cardinal CdLS features such as synophrys, long eyelashes, long philtrum, and thin upper lip vermilion with downturned corners of the mouth (Table 1). Their clinical manifestations and clinical score suggested a classical CdLS.

Among the 11 genes associated with CdLS, we only found potential CdLS causing mutations in the *NIPBL* gene in the three patients. The *NIPBL* gene is located at position 13.2 in the short arm of chromosome 5 (Figure 2a). Delangin protein encoded by *NIPBL* is an ortholog of Scc2 protein in *Drosophila* [1,11] and consists of 47 exons which are anticipated to construct six isoforms (Figure 3a) [20]. The *NIPBL* mutations are reported to be the pathogenic cause in approximately 70–80% of CdLS cases [3,13]. Three de novo heterozygous *NIPBL* mutations have been identified in our patients, which is consistent with the previous studies, in which 99% of *NIPBL*-related CdLS cases are caused by de novo autosomal heterozygous pathogenic mutations [1,9,21].

The missense mutation c.6697G>A in patient CDL1 substitutes valine to methionine at residue 2233. The mutation c.6697G>A is located within the H4 domains of the HEAT-repeat (Huntingtin, elongation factor 3 (EF3), protein phosphate 2A (PP2A), and the yeast kinase TOR1) region [22] which belongs to a highly conserved region of the gene through evolution (Figure 3a). In yeast, Chao et al. suggested that mutations within the HEAT-repeat region lead to nonproductive cohesion-loader interaction, potentially due to impaired Scc1-Scc2 interaction (in humans, RAD21-NIPBL interaction) [23]. The presence of a mutation located within the HEAT-repeat region of *NIPBL* has been reported in CdLS patients with limb defections [9,24]. Consistent with the previous studies, our patient CDL1 also had several limb defects such as clinodactyly of the fifth fingers, partial syndactyly of the left second and third toes, and smallness of hands and feet with limited elbow extension.

In patient CDL2, a deletion mutation c.4504delG in exon 21 of the *NIPBL* gene causes a frameshift that truncates the protein 87 amino acids downstream with a stop codon at residue 1588. This mutation has not been reported in the previous publications. To our best knowledge, only three mutations have been reported in exon 21, including a nonsense somatic mosaic mutation c.4543G>T (p.Glu1515X) [25], a de novo single nucleotide substitution c.4422G4T (p.Arg1474Ser) which could either be missense or affect a donor splice site [18], and a de novo frameshift mutation c.4556_4560del which truncates protein one amino acid downstream [26]. The mutation c.4504delG (p.Val1502TyrfsX87) is the fourth mutation in exon 21 of the *NIPBL* gene.

In the case of patient CDL3, the stop-gain mutation c.2602C>T in exon 10 caused an early stop codon at amino acid 868. Exon 10 which is approximately 8 times larger than the average exon size in *NIPBL* [26], has been known to contain the most mutations in the whole *NIPBL* gene, including two missense, eleven nonsense, twenty-four deletion, and twelve insertion mutations [27]. Phenotypes associated with the reported mutations have not been published in detail, however the correlation of phenotype–genotype was anticipated and the phenotypes of our mutations are consisted with the trend that was reported.

Stop gain, splicing, and frameshift mutations often result in loss-of-function alleles in *NIPBL* due to the deletion of pivotal protein parts [26]. Mutated delangin protein by frameshift mutation c.4504delG in the case of patient CDL2 would exclude a large portion of the amino acid sequence which covers the HEAT repeat regions and the HDAC1 and 3 interacting domains [28]. In patient 3, the stop gain mutation may lead to lack the nuclear localization signal, the part that regulates the transportation of the protein into the nucleus (Figure 3a). Mild phenotypes often presented in patients with deletion of one or a few exons except for the HDAC1 and 3 interacting domains (residues 1838–2000) [27]. The deletion of HDAC1 and 3 interacting domains often results in a more severe CdLS phenotype due to the debilitation of interaction between delangin and other proteins [28]. Consistent with all of the mentioned reports, we noticed a more severe growth delay such as a weight below −3.5 SD and −5.3 SD in patients CDL2 and CDL3 respectively compared to a milder growth delay with a weight below −3 SD in patient CDL1. A greater number of cardinal CdLS features were also recorded in patients CDL2 (4 features) and CDL3 (4 features) compared to patient CDL1 (3 features). Interestingly, while patients CDL2 and CDL3 both have lost-of-function mutations, the cardinal CdLS phenotype in patient CDL2 is slightly milder. However, the overall phenotype of CDL2 is more severe than CDL3 due to a greater degree of major system malfunction such as psychomotor retardation, congenital heart disease, micropenis, abnormal penis, and a notable neurosensory disorder such as epilepsy. In this way, the clinical score criteria suggested by Kline et al. [3] are not associated with the severity of the disease and should be combined with results of molecular testing in order to give a comprehensive diagnostic of CdLS cases. All the above facts suggest a potential pathogenic effect of the identified mutations in our patients.

## 4. Conclusions

This study describes the first whole exome sequencing of Vietnamese patients with CdLS. Whole exome sequencing is an efficient tool for genetic diagnosis of CdLS. The mutations in the *NIPBL* gene detected in this study could be beneficial for further research of CdLS in Vietnamese populations as well as supporting the clinical testing and diagnosis of CdLS in both prenatal and postnatal screening assays. In addition, our results certainly support the conclusion that *NIPBL* mutations are the main cause of CdLS, as found in previous studies.

## Figures and Tables

**Figure 1 medicina-56-00076-f001:**
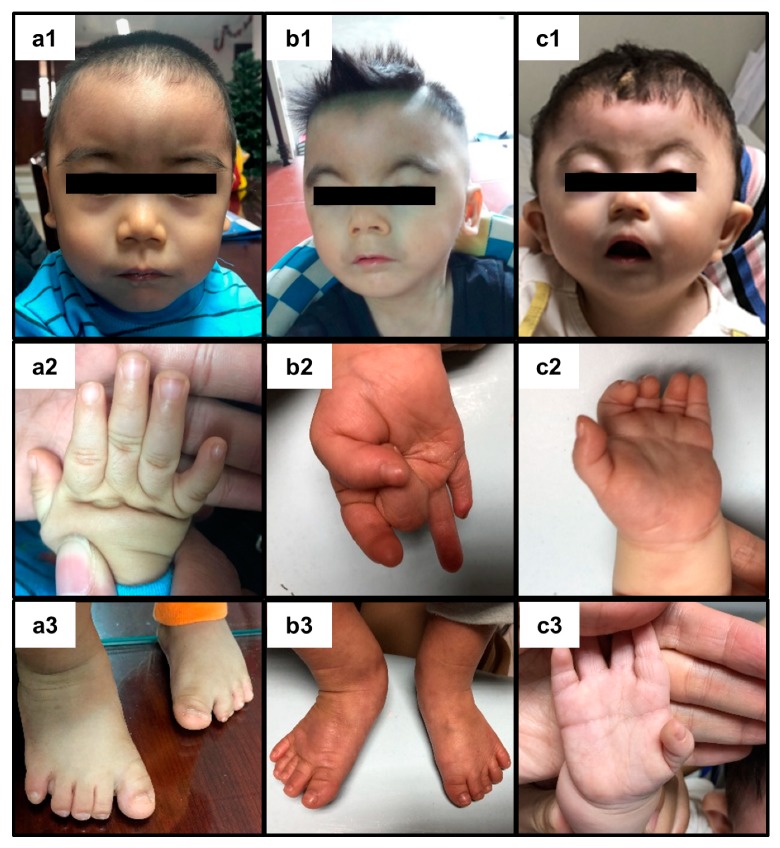
Clinical signs of Cornelia de Lange patients. (**a1**) Dysmorphic face of patient CDL1 with confluent eyebrows, long eyelashes, long philtrum, thin upper lip vermilion, downturned corners of the mouth, low anterior hairline, and low-set ears. Patient CDL1 exhibits a smallness of hands (**a2**) and feet (**a3**) with clinodactyly of the fifth fingers and partial syndactyly of the left and third toes. (**b1**) Photo of patient CDL2 showed arched eyebrows, long eyelashes, short nose with concave nasal ridge and upturned nasal tip, long philtrum, thin upper lip vermilion, and downturned corners of the mouth. (**b2**) Short fifth finger of the right hand of patient CDL2. (**b3**) Small feet of patient CDL2. (**c1**) Patient 3 exhibits arched eyebrows, long eyelashes, short nose with concave nasal ridge and upturned nasal tip, long philtrum, thin upper lip vermilion, and downturned corners of the mouth. **(c2**) Abnormal palmar crease present on the left hand of patient CDL3. (**c3**) Curved fifth finger on the right hand of patient CDL3. The eyes of all patients are covered due to the privacy.

**Figure 2 medicina-56-00076-f002:**
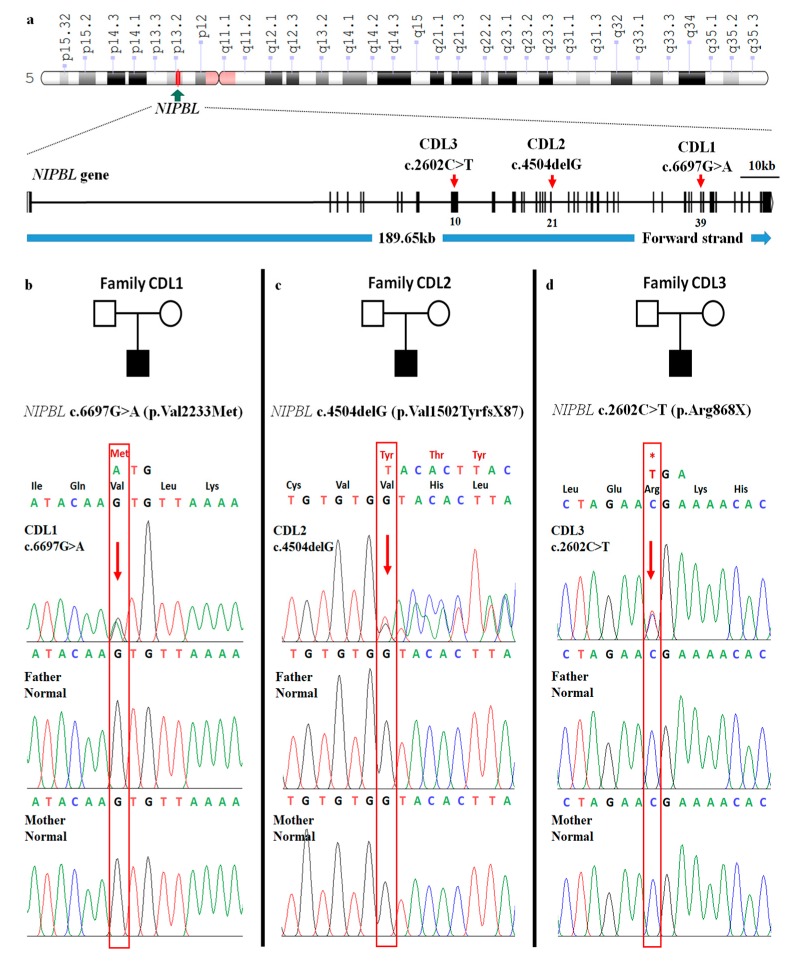
Mapping of the three identified mutations to the *NIPBL* gene. (**a**) A notation of chromosome 5 of the human genome constructed by Genome Decoration Page. *NIBPL* gene is located at position 13.2 on the short arm of chromosome 5. Exon–intron graph of *NIPBL* gene. Among 47 exons of *NIPBL*, the three exons contain identified mutations of CDL1, CDL2, and CDL3 patients were pointed out by arrows. The scale represents the length of 10 kb. (**b**–**d**) Mutations in *NIPBL* in the three CdLS families. Each family has one related case of CdLS. The chromatogram panels show the heterozygous mutation in patients and the normal sequence in the clinically normal parents. Alterations of nucleotides and amino acids (red) are presented above the normal nucleotide and amino acid (black). Mutations included alterations of valine (Val) to methionine (Met) at codon 2233 (Individual II:1, family CDL1) (**b**); a single–base deletion, c.4504delG of exon 25 (Individual II:1, family CDL2) results in premature protein termination after 87 amino acids downstream (**c**); and c.2602C>T (p.Arg868X) (Individual II:1, family CDL3)—an alteration of arginine (Arg) to stop codon at codon 868 leads to the early termination of protein (**d**).

**Figure 3 medicina-56-00076-f003:**
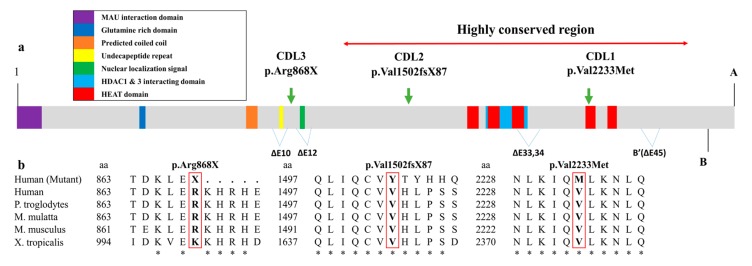
In silico study of the NIPBL protein. (**a**) Schematic illustration of NIPBL protein. The main domains are indicated, including the MAU interaction domain, glutamine rich domain, predicted coiled coil, undecapeptide repeat, nuclear localization signal, HDAC1 and 3 interacting domain, and HEAT domains. The HEAT region contains five domains: H1 (17671805), H2 (1,843–1881), H3 (1945–1984), H4 (2227–2267), and H5 (2313–2351). The NIPBL protein has six isoforms, including A (exon 1–47), B (exon 1–46), ∆E10 (deletion of exon 10 on isoform A), ∆E12 (deletion of exon 12 on isoform A), ∆E3334 (deletion of exon 33 and 34 on isoform A), and B’ (deletion of exon 45 on isoform B). A highly conserved region is located from residue 1150 to residue 2650. (**b**) Conservation of the amino acid changed by each mutation across different species. Alignment of NIPBL protein sequences of human (mutant), human, chimpanzee (*Pan troglodytes*)*,* rhesus monkey (*Macaca mulatta*)*,* house mouse (*Mus musculus*), and western clawed frog (*Xenopus tropicalis*). Identical residues are indicated with (*) below. The positions of the amino acid changes are indicated with a red outline.

**Table 1 medicina-56-00076-t001:** Clinical scores based on clinical presentation of three CdLS Vietnamese patients and molecular genetic analyses.

	CDL1	CDL2	CDL3
**Clinical Score Based on Clinical Presentation**			
**Cardinal features (2 points each if present)**			
Synophrys	2	2	2
Short nose, concave nasal ridge and/or upturned nasal tip	0	2	2
Long and/or smooth philtrum	2	2	2
Thin upper lip vermilion and/or downturned corners of the mouth	2	2	2
Hand oligodactyly and/or adactyly	0	0	0
Congenital diaphragmatic hernia	0	0	0
**Suggestive features (1 point each if present)**			
Global developmental delay and/or intellectual disability	1	1	1
Prenatal growth retardation (<2 SD)	0	0	0
Postnatal growth retardation (<2 SD)	1	1	1
Microcephaly (prenatally and/or postnatally)	1	0	1
Small hands and/or feet	1	1	1
Short fifth finger	1	1	1
Hirsutism	1	0	0
**Total clinical score**	**12**	**12**	**13**
**Molecular genetic analyses**			
**Gene**	*NIPBL*	*NIPBL*	*NIPBL*
**Exon**	39/47	21/47	10/47
**c.DNA mutation**	c.6697G>A	c.4504delG	c.2602C>T
**Protein change**	p.Val2233Met	p.Val1502TyrfsX87	p.Arg868X
**SIFT**	0.002	_	0
**PolyPhen-2**	0.999	_	0.988
**Zygosity**	HET	HET	HET
**Effect**	missense	frameshift	stop-gained
**ClinVar database**	VCV000644755.1	_	VCV000096337.2
**dbSNP database**	_	Novel	rs398124466

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
