# Peer review of "De novo NIPBL Mutations in Vietnamese Patients with Cornelia de Lange Syndrome"

_medicina, 2020, doi:10.3390/medicina56020076_

Round 1

Reviewer 1 Report

The case report presented by Thanh Duong-Chi is interesting because it broadens our knowledge both at the molecular level, a new mutation in the NIPBL gene is described, as at the phenotypic level, because is described this rare syndrome in patients from Southeast Asia. It is a case report with a classic and well built scheme.

Minor revisions:

The clinic of the syndrome in the abstract is repeated twice, the first at the beginning and the second when describing the patients. On line 14, 43 and 166 it says “in which mutations in the NIPBL gene have been accounted for 60% of the cases”. According to two relatively recent reviews, the percentage of mutations in NIPBL in patients with SCdL would be between 70% (Nat Rev Genet. 2018 Oct; 19 (10): 649-666) and 80% (Eur J Hum Genet. 2015 Oct; 23 (10)), I would advise using these values. Reference 8 of the bibliography has nothing to do with the degree of prevalence of the CdLS, nor with the subject of the article. On line 47 instead of HDAC you have to put HDAC8, you must also delete SMC8. On line 49 we speak of -form a rectangle-like structure- referring to the cohesin ring, this is very doubtful. In line 161 of the discussion it is said “Among six genes associated with CdLS”, at present it is considered that there are at least 7 causal genes (Nat Rev Genet. 2018 Oct; 19 (10): 649-666) In line 164-165 of the discussion it is said: “and consist of 47 exons which is anticipated to construct two isoforms of 2804 or 2697 amino acids (Figure 3a).” It should be updated, currently are described at least 6 isoforms (Int J Mol Sci. 2014 Jun 10; 15 (6): 10350-64).  

Reviewer 2 Report

The manuscript by Duong-Chi et al. reports three variants in NIPBL in Cornelia de Lange patients. The content is not novel. The English is very poor and would need careful revision by native English speaker (including description of clinical features). The methods are not described. References are not correctly reported (for example a “30” appears in the text ma not in the bibliography). Further, line 167: “the presence….germline mosaicism”, this sentence is wrong. Why should it be germline mosaicism???? Line 177:”… genetic heterogeneity…” again wrong. Genetic heterogeneity is a description of something completely different from heterozygosity.

Round 2

Reviewer 2 Report

The manuscript has improved. I still believe that language should be checked by a mother tongue (singular/plural, tenses etc).